# The Relationship between Physical Activity Levels and Mental Health in Individuals with Spinal Cord Injury in South Korea

**DOI:** 10.3390/ijerph17124423

**Published:** 2020-06-19

**Authors:** Dong-il Kim, Jeongmin Lee, Hyuna Park, Justin Y. Jeon

**Affiliations:** 1Division of Health and Kinesiology, Incheon National University, Incheon 22012, Korea; dikim@inu.ac.kr; 2Department of Sport Industry Studies, Exercise Medicine and Rehabilitation Laboratory, Yonsei University, Seoul 03722, Korea; leejeo13@gmail.com (J.L.); hyunapark@yonsei.ac.kr (H.P.); 3Exercise Medicine Center for Diabetes and Cancer Patients, ICONS, Yonsei University, Seoul 03722, Korea

**Keywords:** spinal cord injury, physical activity, mental health

## Abstract

*Background:* The aim of this study was to assess the relationship between physical activity (PA) levels and mental health in individuals with spinal cord injury (SCI). *Methods:* Three hospitals in the Seoul metropolitan area were invited to recruit patients with SCI (*n* = 103). PA levels were measured by the Leisure Score Index of the Godin Leisure-Time Exercise Questionnaire (GLTEQ). The Patient Health Questionnaire-9 (PHQ-9), the Generalized Anxiety Disorder-7 (GAD-7) questionnaire, and Multidimensional Scale of Perceived Social Support (MSPSS) were used to assess mental health. *Results:* Compared to the least physically active participants (1st tertile, 44.09 ± 52.74 min/week), the most physically active participants (3rd tertile, 670.86 ± 354.97 min/week) scored significantly lower on PHQ-9 (17.03 ± 5.70 vs. 12.49 ± 4.01, *p* < 0.001), GAD-7 (13.24 ± 4.78 vs. 9.86 ± 3.15, *p* < 0.001), while significantly higher MSPSS (51.24 ± 10.17 vs. 61.37 ± 11.90, *p* < 0.001) after the results were adjusted for age, gender, American Spinal Cord Injury Association impairment scale, and impaired spinal cord levels. Multivariate linear regression analysis showed that the PA was a significant predictor of depression (β = −1.50, *p* = 0.01), anxiety (β = −1.12, *p* = 0.02), and social support (β = 4.04, *p* = 0.01). *Conclusion:* Higher PA participation was associated with lower depression, anxiety, and higher social support scores.

## 1. Introduction

Due to advanced medical care and improved technologies, the life expectancy of people with spinal cord injury (SCI) has considerably increased in recent decades [1]. However, persons with SCI still face major life events that lead to serious physical disability and a higher risk of developing secondary health complications such as cardiovascular disease, insulin resistance, diabetes mellitus, and obesity [2,3,4], which impacts their quality of life (QOL). Therefore, regaining optimal mobility and independence is influenced by biological factors, physical rehabilitation, as well as their QOL and well-being.

Besides an increased risk of developing secondary health complications, people with SCI also have significantly elevated levels of fatigue, anxiety, and depression [5,6,7]. A recent meta-analysis study indicated that the prevalence of depression after SCI is substantially greater than that in the general medical population [7]. Consequently, people with SCI have a higher prevalence of psychological morbidity and suicide [8,9]. Since depressions among people with SCI are higher [10], the suicide mortality rate among people with SCI is nearly five times greater than the general population suicide rate [11]. In addition, according to the Korea Spinal Cord Injury Association (KSCIA), 68.8% persons with SCI had suicidal thoughts, associated with burnout experienced in life, physical disability, and psychiatric symptoms [12].

There is substantial evidence to examine the association between physical activity (PA) participation with health-related fitness level and cardiometabolic health of people with SCI [13,14,15,16]. Recently, the potential role of PA participation in improving psychological well-being and life satisfaction has been recognized along with physical benefit in people with SCI [17,18]. Especially, Naghtingale et al. [17] reported that people with SCI who participated in a moderate-intensity arm-crank exercise program for six weeks showed improved indices of Health-related Quality of life (HRQOL) and reduced fatigue. However, there are relatively fewer studies which investigated the relationship between the amount of PA participation and psychological health of people with SCI.

Social support is defined as an exchange of resources between individuals intended to enhance the well-being of the recipient [19]. Social support plays an important role in the adjustment process of persons with SCI. Social support is defined as an exchange of resources between individuals in social network suggested to improve overall well-being. A systematic literature review shows that social support is related to better physical health, lower pain, effective coping, better adjustment to disability, and higher life-satisfaction, and QOL in persons with SCI [20]. Given the potential role of social support in various positive benefits for persons with SCI, the potential role between social support and PA in people with SCI cannot be underestimated. Previous research demonstrated the positive correlation between social support and PA among several different groups, such as multiple sclerosis [21], cancer survivors [22], and arthritis [23]. Despite these positive findings, there have been only few studies that have investigated the association between participating in PA, psychological health, and the well-being of individuals with SCI.

Therefore, the purpose of this study was to investigate the association between PA participation levels and psychological health variables, including depression, anxiety, and social support in people with SCI.

## 2. Materials and Methods

### 2.1. Participants

This study was approved by the Institutional Review Board of the Yonsei University of Korea (Ethical code: 1040917-201404-HRBR-152-03). Eligible patients were identified from three Seoul metropolitan area hospitals registry on the day before their appointment. While patients were in the waiting room, the research staff explained the purpose of the study to them. Patients who were agreed to participate in the study signed the informed consent and were then interviewed individually by well-trained research staff. One-hundred and three men and women (aged 18–65 years) with SCI of greater than one-year duration (1–48 years) were identified in the study.

### 2.2. Data Collection

The data were collected by research staff using self-reported questionnaires, face-to-face interviews, or objective measurements. In brief, demographic, mental health, and PA data were obtained using self-reported questionnaires. SCI-related medical profiles were collected through face-to-face interviews conducted by research staff. Body Mass Index (BMI) was objectively measured using weight/height measures.

### 2.3. Physical Activity

PA levels were measured by the Leisure Score Index (LSI) of the Godin Leisure-Time Exercise Questionnaire (GLTEQ) [24,25]. The GLTEQ contains three questions that assess the average frequency and duration of mild (minimal effort, no perspiration), moderate (not exhausting, light perspiration), and strenuous (heart beats rapidly, sweating) intensity physical activity for more than 15 min during free time in a typical week. To focus on the PA participation levels as a potential predictor of mental health variables, participants were classified separately into three tertile groups (1st = low PA, 2nd = middle PA, and 3rd = highest PA) according to their total PA participation level. The total time spent in PA per week was calculated by adding the time spent in mild, moderate, and strenuous intensity physical activity per week.

### 2.4. Mental Health

#### 2.4.1. Depression

Depression was measured using the Patient Health Quesionnaire-9 (PHQ-9). The PHQ-9 [26] is a nine-item self-report measure to assess the severity of depression. On each of the nine items, participants are asked to self-rate how often they experienced the indicated symptoms of depression over the previous two weeks on a four-point Likert scale: 0 = “not at all”, 1 = “several days”, 2 = “more than half the days” and 3 = “nearly every day”. The scores on each measure are summed, resulting in a total score range from 0 to 27, with higher scores reflecting severe symptoms of depression (0 to 14 = minimal, 15 to 19 = moderate, and 20 to 27 = severe). This measure demonstrates excellent psychometrics, with good internal consistency (Cronbach α = 0.89) and high test-retest reliability (*r* = 0.84) [26] and has been used extensively in populations with SCI [27,28].

#### 2.4.2. Anxiety

Anxiety was measured by the Generalized Anxiety Disorder-7 (GAD-7) [29]. The GAD-7 is a seven-item self-report questionnaire for measuring the severity of generalized anxiety disorder. On each of the seven items, participants were asked to self-rate how persistent the symptoms were in the past two weeks using a four-point Likert scale: 0 = “not at all”, 1 = “several days”, 2 = “more than half the days” and 3 = “nearly every day”. The scores on each measure are summed, resulting in a total score range from 0 to 21, with higher scores reflecting severe symptoms of anxiety (0 to 4 = minimal, 5 to 9 = mild, 10 to 14 = moderate, and 15 to 21 = severe). The GAD-7 has been validated as a brief screen of anxious symptomatology (α  =  0.89) [30].

#### 2.4.3. Social Support

Perceived social support was measured by use of the Multidimensional Scale of Perceived Social Support (MSPSS). The MSPSS is twelve-item self-report questionnaires evaluate the presence of support from three different categories including family, friends, and others [31]. Each of the items are rated on a seven-point Likert scale from 1 (very strongly disagree) to 7 (very strongly agree) with possible total scores ranging from 12 to 84. Higher scores indicated higher perceptions of social support, and lower scores indicated lower perceptions of social support. The MSPSS demonstrated a very good internal reliability scale (Cronbach’s αs from 0.85 to 0.91), and strong test-retest stability over a two to three-month interval (*r* = from 0.72 to 0.85) [32].

### 2.5. Covariates

Demographic variables included age, gender and BMI. BMI was calculated using measured weight and height (BMI = weight (kg)/height (m)^2^). The severity of the spinal cord injury was determined by the level of neurologic injury and the completeness of injury by the American Spinal Injury Association (ASIA) impairment scale (AIS) [33]. According to AIS, the classification of SCI severity is: A (complete)—no motor or sensory function is preserved in the sacral segments S4–S5, B (incomplete)—sensory function preserved but not motor function is preserved below the neurological level and includes the sacral segments S4–S5, C (incomplete)—motor function is preserved below the neurological level, and more than half of key muscles below the neurological level have a muscle grade less than 3, D (incomplete)—Motor function is preserved below the neurological level, and at least half of the key muscles below the neurological level have a muscle grade of 3 or more.

### 2.6. Data Analysis

All Statistical analysis was performed using SPSS, Windows version 21.0 (SPSS Inc, Chicago, IL, USA). Descriptive analyses are presented for the demographic and PA participation levels, as well as mental health variables were analyzed using weighted means ± standard deviation (SD) for continuous data and numbers (weighted percentages) for categorical data. Complex sample general linear models (CSGLM) were conducted to examine the associations between PA participation levels and each mental health variables. Multiple regression analysis was performed to determine whether there was any association between PA and mental health variables on depression, anxiety, and perceived social support. All tests of statistical significance were two-sided with a *p*-Value of 0.05.

## 3. Results

Demographic, medical-characteristics, and the levels of PA in participants with SCI are descried in Table 1. In brief, 88.3% were men, the mean age of 36.71 ± 9.77 years old, the mean BMI was 21.63 ± 2.97, and 56.3% were of grade AIS-A. Cervical spinal cord injury was most prevalent (57.3%) followed by thoracic (39.0%) and lumbosacral (5.0%). In terms of PA intensities, the average time spent in mild PA was 164.6 ± 217.9 min/week, moderate PA was 115.5 ± 144.9 min per week, and strenuous PA was 47.9 ± 107.9 min per week among participants with SCI. The mean (SD) of the total PA level was 44.1 ± 52.7 min per week in the 1st group (low PA), 239.3 ± 61.3 min per week in the 2nd group (middle PA), and 670.86 ± 354.97 min per week in the 3rd group (highest PA).

Table 2 presents the association between PA participation levels and mental health variables among people with SCI adjusted for age, gender, AIS grade, and impaired spinal cord levels. Participants in the low PA group (1st tertile), mean score of PHQ-9 (17.03 ± 5.70 vs. 12.49 ± 4.01, *p* < 0.001), and GAD-7 (13.24 ± 4.78 vs. 9.86 ± 3.15, *p* < 0.001) was significantly higher than in the group with the highest level of PA (3rd tertile). In addition, participants in the group with the highest level of PA (3rd tertile) had a significantly higher score on MSPSS (*p* < 0.001) than participants with low and medium PA groups (1st and 2nd tertile). 

Multivariate linear regression analysis predicting depression, anxiety, and social support among participants with SCI are shown in Table 3. The model was significant after including age, gender, AIS grade, impaired spinal cord levels, and PA participation in depression (R^2^ = 0.286; F-value = 7.755, *p* < 0.001), anxiety (R^2^ = 0.252; F-value = 6.545, *p* = 0.001), and social support (R^2^ = 0.171; F-value = 4.003, *p* = 0.002). In each model, the PA participation is a significant predictor of depression, anxiety, and social support (*p* < 0.05). 

## 4. Discussion

In our study, we examined the association between PA participation levels and mental health variables, including depression, anxiety, and perceived social support in Korean people with SCI. Our results indicate that the PA was favorably associated with depression, anxiety, and perceived social support after adjusting covariates. Specifically, we found that the more PA was associated with a lower scale value of depression and anxiety and a higher scale value in perceived social support.

Several epidemiological studies have investigated the association between PA and psychiatric comorbidities such as depression and anxiety in people with SCI [34,35]. For instance, one cross-sectional study on 169 Japanese male individuals with SCI reported that individuals who had participated in sports activity more than three times a week (high-active group) had lowest scores of depression and anxiety compared to middle-active group (once or twice a week), low-active group (once to three times a month), and inactive group (no sports participation) [34]. However, PA participation levels between the groups in this study were measured with multiuse dichotomized variables by frequencies of sports activity participation per week. Another cohort study with 137 Italian men with SCI reported that, compared to the physically inactive individuals with SCI, those who were highly physically active had lower scores of depression and anxiety [35]. This study was limited; however, where the use of the PA participation levels between the groups was measured by dichotomized variables as “high frequency of physical activity (three times a week)” or “no PA participation” but it did not examine the estimated number of minutes per week spent on PA. Together, most findings from these studies showed that there are significant associations between PA and psychiatric comorbidities such as depression and anxiety in people with SCI. These studies, however, were limited in terms of the estimated amount of time spent in PA and types of intensity. Our study adds to these finding by examining the association between comorbid psychological conditions (depression and anxiety) by the estimated number of minutes per week spent in PA (low PA vs. middle PA vs. highest PA). Social support is positively influenced in PA participation for people with SCI [20]. Loy et al. [36] reported that social support was a significant predictor (*p* < 0.05) of leisure participation for people with SCI. Another population-based cohort study by Ginis et al. [37] reported that greater social integration was associated with a greater likelihood of being physically active. In accordance with these findings, our results showed that participants in the 3^rd^ tertile (the most active) had significantly higher scores on social support (MSPSS, *p* < 0.001) than participants in the 1st (the least active) and the 2nd tertile. These results are in accordance with well-documented evidence which showed that perceived social support reflect the total PA participation level [38]. The psychological mechanisms underlying the association between social support and PA are not fully understood. Social support may be a powerful way of fostering healthy behavior [20,39], which may subsequently play a role in weight management and engaging exercise among people with SCI. This study contributes to the existing literature in several ways. Unlike previous studies, including a single intensity or measure of PA by dichotomous variables, the current study measured each different intensity-specific PA and estimated amount of PA min per week among participants with SCI in Korea. In addition, by including depression, anxiety, and social support as outcomes, this study expands our understanding of the association between PA and psychological well-being among people with SCI. However, our study has several limitations. First, this current study was relatively limited in the number of participants. Therefore, the generalization of the results needs caution. Second, the PA questionnaire used in this study had potential biases in our participants. In particular, PA levels using the self-reported measure could be more likely to be overestimated compared to the actual time spent in PA [40]. In our study, the total PA was over 300 min per week, which was substantially higher than the American College of Sport Medicine PA guidelines of ≥150 min of moderate to vigorous PA per week [41]. Third, our study did not conduct subgroup analysis according to gender due to the small sample size of female participants with SCI. It is possible that the relationship between PA and mental health may vary by gender; consequently, future studies should obtain larger sample sizes and stratify the analysis by gender. Fourth, the intensity of the PA was not reflected because PA participation levels were calculated by adding the min per week spent with different types of intensity. Last, it should be noted that the findings of this study may not be generalizable to people with SCI in other countries due to the unique distribution of SCI levels as well as differences in cultural, socioeconomic, and environmental/lifestyle factors in Korean people with SCI.

## 5. Conclusions

In summary, we found that there were significant associations between PA, depression, and anxiety and perceived social support in Korean people with SCI. Specifically, more physically active participants with SCI had lower scores of depression and anxiety. Moreover, given that many people with SCI experience challenges in their activities of daily living which result in decreased QOL, the relationship between social support and PA participation among people with SCI is of interest. The findings of our study prompt future longitudinal studies and clinical trials to further investigate the effects of PA on mental health in Korean people with SCI. In addition, it may be important for health care providers in clinical settings to encourage people with SCI to participate in sufficient levels of PA for potential mental health benefits. Regardless, further research using a more rigorous study design (i.e., experimental, longitudinal) is required to build on our findings.

## Figures and Tables

**Table 1 ijerph-17-04423-t001:** Characteristics and physical activity profile of participants with spinal cord injury.

Variables	Total	1st T	2nd T	3rd T
	(*n* = 103)	(*n* = 33)	(*n* = 35)	(*n* = 35)
Male	91 (88.3)	25 (75.8)	33 (94.3)	33 (94.3)
Age (years)	36.71 ± 9.77	38.94 ± 9.87	36.43 ± 9.88	34.89 ± 9.40
Height (cm)	172.91 ± 11.63	172.18 ± 7.41	174.03 ± 5.26	172.49 ± 18.02
Weight (kg)	65.37 ± 10.48	63.85 ± 9.03	67.11 ± 8.57	65.06 ± 13.17
BMI (kg/m^2^)	21.63 ± 2.97	21.50 ± 2.47	22.16 ± 2.61	21.23 ± 3.67
AIS grade				
A	58 (56.3)	20 (60.6)	18 (51.4)	20 (57.1)
B	36 (35.0)	10 (30.3)	13 (37.1)	13 (37.1)
C	7 (6.8)	2 (6.1)	4 (11.4)	1 (2.9)
D	2 (1.9)	1 (3.0)	0 (0.0)	1 (2.9)
Impaired spinal cord levels				
Cervical	59 (57.3)	22 (66.7)	19 (54.3)	18 (51.4)
Thoracic	39 (39.0)	9 (27.3)	15 (42.9)	15(42.9)
Lumbosacral	5 (5.0)	2 (6.1)	1 (2.9)	2 (5.7)
PA intensity				
Mild (min/week)	164.6 ± 217.9	32.6 ± 49.5	116.8 ± 78.2	335.7 ± 289.1
Moderate (min/week)	115.5 ± 144.9	11.5 ± 29.7	94.6 ± 55.8	233.4 ± 181.1
Strenuous (min/week)	47.9 ± 107.9	-	37.8 ± 58.2	101.7 ± 160.4
Total PA (min/week)	323.4 ± 336.5	44.1 ± 52.7	239.3 ± 61.3	670.9 ± 354.9

Values are mean ± SD or *n* (%). Abbreviation: T = tertiles; AIS = American Spinal Cord Injury Association impairment scale, A = (complete)—no motor or sensory function is preserved in the sacral segments S4–S5, B = (incomplete)—sensory function preserved but motor function is not preserved below the neurological level and includes the sacral segments S4–S5, C = (incomplete)—motor function is preserved below the neurological level, and more than half of the key muscles below the neurological level have a muscle grade less than 3, D = (incomplete)—motor function is preserved below the neurological level, and at least half of the key muscles below the neurological level have a muscle grade of 3 or more; PA = physical activity. Physical activity was categorized into three tertile groups: (1) low PA (1st T), (2) middle PA (2nd T), or (3) highest PA (3rd T). Total physical activity = (min of mild physical activity per week) + (min of moderate physical activity per week) + (min of strenuous physical activity per week).

**Table 2 ijerph-17-04423-t002:** Association between physical activity and mental health variables in participants with SCI.

Variables	Physical Activity Participation			*p* for Trend
	1st T (*n* = 33)	2nd T (*n* = 35)	3rd T (*n* = 35)	
Depression (PHQ-9)	17.03 ± 5.70	14.71 ± 4.79	12.49 ± 4.01 *	*<0.001*
Anxiety (GAD-7)	13.24 ± 4.78	10.86 ± 3.19 *	9.86 ± 3.15 *	*<0.001*
Social Support (MSPSS)	51.24 ± 10.17	54.49 ± 11.74	61.37 ± 11.90 * †	*<0.001*

Values are mean ± SD. Complex sample general linear model analysis adjusted for age, gender, American spinal cord injury association impairment scale and impaired spinal cord levels. *P*-values in italics indicate a statistical significance of α < 0.05. Abbreviation: T = tertile; PHQ-9 = Patient Heath Questionnaire-9; GAD-7 = Generalized Anxiety Disorder-7. Physical activity was categorized into three tertile group: (1) low PA (1st T), (2) middle PA (2nd T), or (3) highest PA (3rd T); * significantly different from the “1st tertile” group in physical activity. † significantly different from the “2nd tertile” group in physical activity.

**Table 3 ijerph-17-04423-t003:** Multivariate linear regression analysis predicting depression, anxiety, and social support.

Dependent Variable	Predictor	B	SE	*β*	95% CI for B (Lower, Upper)	*p*-Value
**Depression (PHQ-9)**	Age	0.12 *	0.05	0.23 *	0.21, 0.27	0.01
	Gender	5.1 *	1.43	0.32 *	2.22, 7.89	<0.001
	AIS grade	−0.68	0.64	−0.09	0.21, 0.27	0.3
	Impaired spinal cord level	−1.60 *	0.8	−0.18 *	−3.10, −0.05	0.04
	Physical activity Participation	−1.50 *	0.6	−0.23 *	−2.60, −0.31	0.01
**Anxiety (GAD-7)**	Age	0.05	0.04	0.12	−0.24, 0.12	0.18
	Gender	3.86 *	1.13	0.31 *	1.62, 6.1	0.001
	AIS grade	−0.02	0.50	−0.003	−1.01, 0.9	0.97
	Impaired spinal cord level	−1.44 *	0.6	−0.21 *	−2.64, −0.23	0.02
	Physical activity Participation	−1.12 *	0.5	−0.23 *	−2.01, −0.23	0.02
**Social Support (MSPSS)**	Age	−0.19	0.1	−0.15	−0.42, 0.05	0.14
	Gender	−4.71	3.6	−0.13	−11.8, 2.4	0.38
	AIS grade	0.76	1.59	0.05	−2.4, 3.9	0.89
	Impaired spinal cord level	3.16	1.92	0.15	−0.7, 6.9	0.10
	Physical activity Participation	4.04 *	1.43	0.27 *	1.2, 6.9	0.01

Abbreviations: CI = confidence interval, PHQ-9 = Patient Heath Questionnaire-9, GAD-7 = Generalized Anxiety Disorder-7, AIS = American Spinal Cord Injury Association, AIS = American spinal cord injury associations impairment scale, B = Unstandardized coefficient, S.E = Standard Error, *β* = Standardized Coefficient. * *p* < 0.05.

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
