# Peer review of "The Relationship between Physical Activity Levels and Mental Health in Individuals with Spinal Cord Injury in South Korea"

_ijerph, 2020, doi:10.3390/ijerph17124423_

Round 1
Reviewer 1 Report
First at all, congratulations for your work, it is very interesting.
In order to improve the quality of your study, I have attached some comments in the file.
In brief I recommend you to revise your writing, update the references, add effect size in table 2 and odd ratios (95%CI) in table 3.

Author Response
June 7th 2012
Dear Reviewer
Ref.: Manuscript ID ijerph-822381
Thank you very much for reviewing our manuscript. We also greatly appreciate the reviewer for your complimentary comments and suggestions. We have carried out the experiments that the reviewers suggested and revised the manuscript accordingly
Please see the attachment file a point-by-point response to reviewer’s concerns. We hope that you find our responses satisfactory and that the manuscript is now acceptable for publication.
Sincerely,
Justin Y. Jeon
Department of Sport Industry Studies,
Exercise Medicine Center for Diabetes and Cancer Patients,
Yonsei University, Shinchon-dong, Seodaemun-Gu, Seoul, 120-749,South Korea
Tel: (82) 2-2123-6197
E-mail: jjeon@yonsei.ac.kr

Reviewer 2 Report
The manuscript entitled “The relationship between physical activity participation levels and mental health in individuals with spinal cord injury in South Korea” deals with a study of a group of 103 people with spinal cord injury, which was motivated to have physical activity, showing that depression and anxiety were reduced among them, while social support was increased.
In the present form, the study should be considered only as a case report, where the analysis is carried out with tools that are well known, and whose novelty only relies on the application of a physical activity onto a South Korean context.
Specific comments:
Line 27: Define ASIA.
Line 216: Conclusion part is too short. I think the authors found interesting results that must be included in this section.
Author Response
June 7th 2012
Dear Reviewer
Ref.: Manuscript ID ijerph-822381
Thank you very much for reviewing our manuscript. We also greatly appreciate the reviewer for your complimentary comments and suggestions. We have carried out the experiments that the reviewers suggested and revised the manuscript accordingly
Please see the attachment file a point-by-point response to reviewer’s concerns. We hope that you find our responses satisfactory and that the manuscript is now acceptable for publication.
Sincerely,
Justin Y. Jeon
Department of Sport Industry Studies,
Exercise Medicine Center for Diabetes and Cancer Patients,
Yonsei University, Shinchon-dong, Seodaemun-Gu, Seoul, 120-749,South Korea
Tel: () 2-2123-6197
E-mail: jjeon@yonsei.ac.kr

Reviewer 3 Report
I commend the authors on their work entitled, “the relationship between physical activity participation levels and mental health in individuals with spinal cord injury in South Korea.”
I have a few major concerns as well as significant English editing that should be addressed in the following manuscript. Major concerns are listed first.
Item 1: Materials and Methods, Line 72: inclusion criteria item 3 states that individuals who had not exercised regularly in the past 6 months before preceding the study. Then the methods section transitions to how PA levels were measured, however there is no information on an exercise program? It reads as if a description of the exercise program section 2.2 is missing entirely from the materials and methods section.
Materials and Methods, Physical Activity section: The authors should consider reporting the average minutes for each intensity as well as total PA participation as described in Table 1. Moreover, it should be investigated whether certain intensities as well as duration influence the mental health outcomes.
Item 2: Discussion, paragraph 2. There appears to be a substantial gap that the authors haven’t acknowledged in terms of exercise intensity vs. total, estimated PA participation. Although is that mentioned as a limitation, I believe the manuscript would have more value if the authors analyze the relationship of estimated exercise intensity in addition to total PA participation. Once revised accordingly, the title may be to be adjusted as well.
Item 3: Discussion, Lines 191 – 215: different color font (dark gray) compared to the rest of the text. I have pause when I see text in a different color font as it implies material was copied and pasted.
Other edits:
Line 12: correspondence is listed twice
Abstract, Line 23: spacing between 670.86 ± 354.97 min/week
Abstract, Line 25: PA levels were (plural)
Abstract, Line 27: insert [after] adjusted for
Abstract, Line 27: define ASIA - American Spinal Cord Injury Association
Abstract, Line 28-29: higher social support scores (plural)
Keywords, Line 30: uncapitalize support [of social support]
Introduction, Line 33: refine SCI in the introduction
Introduction, Line 34: revise to persons with SCI, in place of person.
Introduction, Line 37: revise to which impacts their quality of life
Introduction, Lines 37-39: challenging phrasing for the reader. Consider revising to the following or phrasing. “Therefore, to regain optimal mobility and independence is influenced by biological factors, physical rehabilitation, as well as their QOL and well-being.”
Introduction, Line 40: revised to besides
Introduction, Line Line 42: Miglorini et al., sentence should be moved to after line 41 [elevated levels of fatigue, anxiety, and depression.] Followed by, consequently, people with SCI have higher prevalence…
Introduction, Line 46: issue with reference citation inserted into the word Ac[cording]
Introduction, Line 26: revise person to persons with SCI (plural)
Introduction, Line 49: revise to “there is ample evidence that PA”
Introduction, Line 53: revise to who participated [in or in an ] exercise program[s] for
Introduction, Line 55: insert investigated [the] relationship
Introduction, Line 60: revise to persons with SCI
Introduction, Line 62: MS citation spacing for reference 29
Introduction, Line 63: revise to few studies that have investigated the association…PA participating, psychological health, and well-being of individuals with SCI
Introduction, Line 66: uncapitalize people
Materials and Methods, Line 71: criteria included:, not ;
Materials and Methods, Line 72: 2) between ages 18 and 65 years old
Materials and Methods, Line 72: 3) define the criteria for not exercising regularly
Materials and Methods, Line 83: revise to in a typical week
Materials and Methods, Lines 83-84: How was intensity assessed? What criteria and methods?
Materials and Methods, Line 87: citation format issue
Materials and Methods, Line 89: tab space needs to be removed for 2.3 mental health
Materials and Methods, Line 90: numbering for depression header does not align with the numbering
Materials and Methods, Line 105: citation issue number 70
Materials and Methods, Line 109: citation issue number 68
Materials and Methods, Line 119: Data Analysis has the same numbering as the mental health header
Materials and Methods, Line 125: consider stating the ASIA classifications (A, B, C, D, E) and neurological level of injury in text.
Results, Line 129: earlier the authors state males and females, here men and women phrasing are used. Stay consistent.
Results, Table 1: E is not listed in the table, list 0 for this category if appropriate
Results, Line 141: ASIA, and level of injury. Insert comma
Results, Line 151: level of injury, and disease
Discussion, Line 182: revise to the group who had participated in PA more than
Discussion, Lines 184, 186: revise to leisure time
Discussion, Line 190: revise to number of minutes
Discussion, Line 197: typographical error SC, instead of SCI
Table 3 supplementary. Spacing after table 3.
Author Response

(The authors gave the same response as above.)

Round 2
Reviewer 3 Report
Overall, the authors did a satisfactory job addressing initial comments and concerns. My minor comments below primarily reflect new edits.
Minor edits:
Materials and Methods, Lines 68-127: The section regarding study participants is satisfactory in the response/cover letter. However, the first sentence in the response document is not included in the revised manuscript, "The survey was conducted at Shinchon Severance Hospital in Seoul, Korea." As the reviewer, I am indifferent whether its included, but wanted to bring it to your attention.
Discussion, Lines 189-193: The Italian SCI study does not appear to be cited within this description. Also, review how they phrased high frequency vs. no PA. The manuscript is cited above 35-37, but I would also recommend citing each paper as you describe it within the discussion.
Discussion, Line 199: 'any by different intensity' may be unintentionally misleading. Although different intensities were estimated and recorded, differences in intensities were not assessed/evaluated. I would suggest expanding and clarifying for the reader.
Additional language edits:
Materials and Methods, Line 84: through in place of though
Materials and Methods, Line 85: I don't recall seeing BMI (body mass index) defined earlier. If not, please revise.
Materials and Methods, Line 88: Your response document states this header is labelled as Physical Activity. Also, you've listed Data Collection as a header twice for both 2.2 and 2.3. Please revise accordingly.
Materials and Methods, Lines 110, 119, 128: Verify that 1), 2), and 3) are the correct sub headers per journal guidelines.
Materials and Method, Line 150: comma after characteristics
Materials and Methods, Line 150: Typographical error descried should be described.
Materials and Methods, Lines 156-157: remove capitalization for low, middle and highest.
Materials and Methods, Table 1: C in characteristics is bolded.
Materials and Methods, Table 1: I would reconsider removing female column from the table since the difference out of 100% from male is implied. For example, if there's 88.3% males, the reader knows there's 11.7% females. Also note, text description above used men and women. Here male/female are used. I would recommend choosing either or men/women vs. males/females.
Materials and Methods, Table 2: do not capitalize 'mental' in table description.
Materials and Methods, Table 3: Nice addition of 95% CI. Consider formatting as you have described it in the table header. (0.27, 0.21), removing "to". Please review 95% CI for: depression - predictor age and AIS grade, and anxiety predictor - gender. These three are not presented in a lower, upper format (could possibly be typographical error?)
Discussion, Line 184: double spacing after with SCI for the references.
Discussion, Line 204: space needed after most active
Discussion, Line 207: Revise "level of PA" to reflect the total PA time (not necessarily reflected in PA intensity. Level of PA infers intensity.
Discussion Line 222: Capitalize organization "American College of Sports Medicine". Also, since ACSM isn't referred to again in the manuscript, I would remove the ACSM abbreviation.
Discussion, Line 224: I could consider expanding on why moderate intensity was reported as mild intensity.
Discussion, Line 225: Typographical error, sized should be size.
Discussion, Line 228-229: It is possible that time spent in certain PA intensities could influence your outcomes.
Overall, great work on the revision. Please review and consider these minor edits.
Author Response
June 11th 2020
Dear Reviewer
Ref.: Manuscript ID ijerph-822381
We thank the reviewer for the second round of comments and the opportunity to further revise our manuscript.
We also greatly appreciate the reviewer for his thoughtful and thorough review and believe his input has been invaluable to make our review more balanced
Please find attached file a point-by-point response to reviewer’s concerns. We hope that you find our responses satisfactory and that the manuscript is now acceptable for publication.
Sincerely,
Justin Y. Jeon
Department of Sport Industry Studies,
Exercise Medicine Center for Diabetes and Cancer Patients,
Yonsei University, Shinchon-dong, Seodaemun-Gu, Seoul, 120-749,South Korea
Tel: () 2-2123-6197
E-mail: jjeon@yonsei.ac.kr
